# Habitat occupancy of the threatened Diademed Plover (*Phegornis mitchellii*) is not affected by llama grazing or peatland size, but declines with peatland humidity

**Alejandro G. Pietrek**[1,2]*, **Kristina L. Cockle**[3], **Andrea E. Izquierdo**[4], **Viviana S. Berrios**[5], **Bruce E. Lyon**[1]

**1** Ecology and Evolutionary Biology Department, University of California Santa Cruz, Santa Cruz, CA, United States of America, **2** Instituto de Bio y Geociencias del NOA (IBIGEO), Universidad Nacional de Salta, Consejo Nacional de Investigaciones Científicas y Técnicas (CONICET), Salta, Argentina, **3** Instituto de Biología Subtropical (IBS), Consejo Nacional de Investigaciones Científicas y Técnicas (CONICET), Puerto Iguazú, Misiones, Argentina, **4** Instituto Multidisciplinario de Biología Vegetal (IMBIV), Consejo Nacional de Investigaciones Científicas y Técnicas (CONICET), Universidad Nacional de Córdoba, Córdoba, Argentina, **5** Facultad de Ciencias Naturales, Universidad Nacional de Salta (UNSa) y Consejo Nacional de Investigaciones Científicas y Técnicas (CONICET), Salta, Argentina

* apietrek@ucsc.edu

## Abstract

Many habitat-specialist organisms occur in distinct, patchy habitat, yet do not occupy all patches, and an important question is why apparently suitable habitat remains unoccupied. We examined factors influencing patch occupancy in near-threatened, little-known Diademed Plovers (*Phegornis mitchellii*), arguably the bird most specialized to life in High Andean peatlands. Andean peatlands are well-suited to occupancy modelling because they are discrete patches of humid habitat within a matrix of high-altitude steppe. We hypothesized that Diademed Plovers occupy preferably larger and more humid peatlands, and avoid peatlands used for grazing by llamas and vicuñas, which may trample vegetation and nests. From December 2021 to February 2022 (breeding season), we conducted plover occupancy surveys (2–4) on 40 peatlands at Lagunas de Vilama, a landscape of arid steppe and wetlands above 4,500 m in NW Argentina. We measured peatland size, grazing pressure, topographic and remotely-sensed variables that correlate with humidity, and incorporated these as covariates in occupancy models. Occupancy models showed that more than 50% of the studied peatlands were used by Diademed Plovers and most showed signs of reproduction, highlighting the importance of the Vilama Wetlands for Diademed Plover conservation. Within peatlands, Diademed Plovers were most often associated with headwaters. The top ranked occupancy model included constant detection, random spatial effects, and a single occupancy covariate: mean NDWI (Normalized Difference Water Index, an index correlated with water content and humidity) over the previous three years. Contrary to our prediction, Diademed Plovers preferred less water-saturated peatlands (lower NDWI), possibly to avoid nest flooding. This may be especially important in wet years, like the year when we conducted our surveys. Neither peatland size nor grazing by llamas and vicuñas affected peatland use by Diademed Plovers, suggesting that llama grazing at current levels

**Data Availability Statement:** All data files and R code are available from my personal and public GitHub database (https://github.com/aptrk).

**Funding:** This work was funded with a grant from the Neotropical Bird Club (https://www.neotropicalbirdclub.org/) awarded to AGP. The sponsors did not play any role in the study design, data collection and analysis, decision to publish or manuscript preparation.

**Competing interests:** The authors have declared no competing interests.

may be compatible with plover conservation. For organisms that specialize on humid habitats, such as peatlands, factors affecting occupancy may vary temporally with variation in climate, and we recommend follow-up surveys across multi-year timescales to untangle the impact of climate on animals' use of humid habitats.

## Introduction

A major aim of population ecology is to identify the factors that determine how species are distributed in space. From a purely biological perspective, knowing where a species lives provides a foundation for understanding habitat requirements, food habits, refuge and interactions with other species, while from an applied perspective, identifying key habitat is a critical step in species conservation. For organisms that occur in distinct, patchy habitat, an important question is why individuals occur in some patches and not in others, when unoccupied patches appear to offer suitable habitat. Patches may be unoccupied for a number of reasons. First, unoccupied patches may be unsuitable because of their abiotic environment or the presence/absence of other species [1, 2]. Second, unoccupied patches may be suitable, but population size may be lower than that needed to saturate all suitable habitat. For example, populations may be regulated by constraints in the non-breeding season, so that breeding density is too low to fill all suitable breeding habitat [3–5]. Third, if suitable habitat varies in quality and is not saturated, mobile organisms may preferentially occupy the best quality habitats [6–9]. In that case, differences between unoccupied and fully occupied habitat should reflect variation in features along a spectrum from unsuitable to high quality habitat.

Peatlands in the High Andes of South America occur as discrete, humid habitat patches along streams within an arid matrix of high-altitude steppe. These peatlands are oases that harbor highly distinct biological communities, including many endemic amphibians and birds [10–12]. Indigenous herders have used and managed these peatlands to graze llamas for millennia [13–15]. However, these biodiversity islands are experiencing rapid changes in vegetation related to declining humidity, caused by aridization [16–18] and escalating water requirements for mining activities [19–21]. Other localized threats, such as overgrazing, pollution and introduction of invasive trout (*Salmo trutta*) may further endanger the biodiversity of some Andean peatlands [22, 23]. To address the consequences of these threats, we need an understanding of how peatland features and human-driven changes affect peatland-dwelling vertebrates.

The Diademed Plover (*Phegornis mitchellii*), a shorebird, is arguably one of the most specialized birds adapted to life in Andean peatlands. The Diademed Plover is categorized as near-threatened by the IUCN and is mostly found in the peatlands of the High Andes, spanning from Southern Peru (above 4,000 m.a.s.l) to Argentina and Chile (where it can be found as low as 2,000 m [24]). Diademed Plovers primarily feed on macro-invertebrates and amphibians, often within streams [25]. They usually occur alone or in pairs, but they are often difficult to find [26, 27], making them a sought-after species among birdwatchers worldwide. These plovers make a rudimentary nest on the ground and females lay two eggs in late spring. Nests are placed on cushion plants or among rocks in rocky patches. Both parents incubate the eggs and provision the precocial chicks. The breeding grounds of Diademed Plovers are often inaccessible to observers during the winter, and winter distribution data are scarce. In areas like Vilama, where many peatlands freeze in winter, plovers might be altitudinal migrants [26]. Importantly, their specific micro-habitat requirements remain largely unknown. Peatlands

used by Diademed Plovers look, to the human eye, very similar to peatlands where they are not found [28]. To conserve habitat for Diademed Plover, it is important to identify the factors that influence their occurrence in peatlands of the High Andes.

For species like the Diademed Plover, using models based on occurrence (presence or presence-absence) can help identify factors affecting occupancy and also provide a cost-effective method to assess population status based on occupancy rather than abundance [29–31]. The beginning of this century has witnessed an explosion of approaches to model large scale occurrence, ranging from presence background models from opportunistic surveys, museum collections or citizen science data (usually encompassing large areas) to site-occupancy models (mostly from planned surveys and relatively small areas). Site-occupancy models are the preferred option for habitat specialists, especially those not found in all patches and occurring at relatively low densities. More recent developments of occupancy models include the modeling of spatial random effects, acknowledging that much of the unmodeled variation is inherently spatial and points closer in space tend to be correlated [32, 33].

Factors that might influence occupancy by Diademed Plovers include peatland area, humidity, and grazing pressure. Patch size has been traditionally linked to higher diversity, population sizes or more generally occurrence [34–36]. For Diademed Plovers, larger peatlands are likely to harbor more prey and greater micro-habitat diversity (vernal pools, running water) for their prey and offer more suitable nesting sites. Due to the Diademed Plovers' primary dependence on aquatic prey, they might be expected to associate with peatlands that are either more humid on average (more aquatic prey), or more hydrologically stable (providing a consistent and predictable food supply throughout the season). However, we still lack empirical evidence that supports this in High Andean peatlands. Overgrazing has also been suggested to have a negative effect on plover occurrence [28], despite these peatlands having a long pastoral tradition. Trampling by llamas (*Lama glama*) and vicuñas (*Vicugna vicugna*) may disturb nesting behavior and habitat, as reported for livestock effects on nesting shorebirds in coastal habitats [37], and the effect of livestock on plover occurrence may be area-dependent (with larger effects on smaller peatlands than larger peatlands). If plovers avoid llamas and vicuñas by using different peatlands, we predict plover occupancy to be negatively associated with number of llamas and vicuñas. If plovers tolerate llamas and vicuñas in the same peatlands (at current densities), we predict that occupancy will not be associated with grazing pressure.

The aim of this paper was to identify peatland features that Diademed Plovers use during their breeding season, when they establish clearly-delimited territories within peatlands [25, 26, 38]. We conducted planned, repeated surveys on 40 peatlands in the High Andes of NW Argentina and used single season occupancy models to explore peatland variables that may affect occurrence of Diademed Plovers. We hypothesized that plovers would respond positively to peatland size and negatively to grazing by llamas and vicuñas. Thus, we predicted that plover occupancy would increase with peatland size and humidity and decline with grazing pressure. Additionally, we explored the locations, within peatlands, used by the Diademed Plovers, to determine whether plovers were distributed randomly or toward one end of these somewhat linear habitats. Specifically, these wetland habitats form around streams that emerge from the sides of mountains, resulting in habitat patches that are typically much longer than wide. We sought to determine if plovers were equally likely to occur along the entire length of these habitat patches or more likely to occur at the tops where the headwaters emerge from the mountains. Our findings aim to contribute not only to the fundamental understanding of Diademed Plovers' biology but also to provide guidance for conservation planning in a region experiencing rapid transformations driven by mining activities and climate change.

## Methods

### Study system

We conducted our study at Lagunas de Vilama in the High Andes of Jujuy Province, NW Argentina near the border of Bolivia and Chile (22° 30″ S, 66° 55″ W) (Fig 1, [39]). Lagunas de Vilama (hereafter Vilama) encompasses numerous lakes and peatlands embedded in an arid matrix above 4,500 m.a.s.l. Vilama is arguably one of the most diverse Andean wetlands systems of Argentina. It has been a RAMSAR site since 2000, and was designated an Important Bird Area (IBA) in 2003 [40, 41]. Geologically, this site is part of the Vilama caldera that collapsed 8.5 million years ago and is surrounded by several peaks and volcanoes that reach near 6,000 m.a.s.l [42]. The climate is classified as arid desert cold [43] with an extreme temperature amplitude, nights with temperatures below freezing throughout the year and high UV radiation. About 80% of precipitation at Vilama is concentrated in the summer-autumn months. There are often inter-annual fluctuations in rainfall, alternating between wet and dry periods [18]. For example, mean annual precipitation between 2008 and 2019 was 177 mm (range: 65–315), and precipitation between July 2021 and June 2022 was 244 mm (38% over the annual mean).

The landscape at Vilama is a grass steppe with a few shrub species (*Adesmia sp.*, *Parastrephia sp.*) at low density [44]. Within this matrix, peatlands occur as linear features found along streams and fed by snowfall in the headwaters. These peatlands are primarily formed by a high

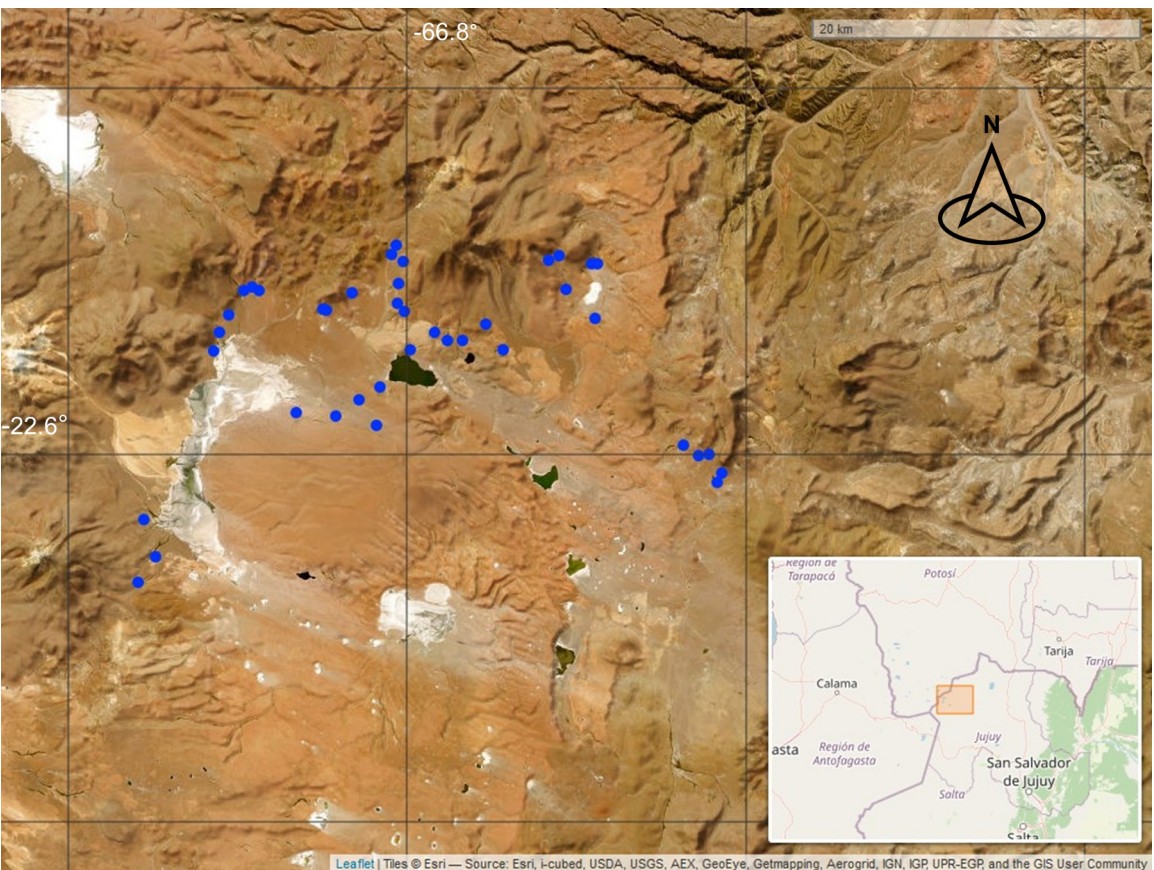

**Fig 1. Vilama wetlands.** Each dot represents a surveyed peatland.

cover of cushion-like plants of the genera *Distichia* and *Oxychloe* (Juncaceae), *Deyeuxia* and *Cinnagrostis* bunch grasses (Poaceae) and sedges of the genus *Phylloscirpus* (Cyperaceae). Peatlands at Vilama can be classified as types 1 and 2 according to Izquierdo et al. [45]: they are generally dominated by cushion plants of the genus *Distichia* and *Oxychloe* and are located in relatively steep areas above 4000 m.a.s.l. Each summer between late November and early February (when most plovers nest), Kolla shepherds from the neighboring community of Lagunillas del Farallón move up to Vilama to graze their llamas on the peatlands. No people live at Vilama in winter. The Kollas comprise a group of tribes that inhabit northwest Argentina, northern Chile, and Bolivia and have held common land property in this area for centuries.

## Peatland mapping and extraction of peatland characteristics

We used a regional map of peatlands described in [45]. This map is a product of a Random Forest supervised classification of Sentinel 2 optical images and had an overall accuracy of 82%. For all peatland polygons within our study area, we calculated five spatial and eight spectral variables on the Google Earth Engine (GEE) platform [46]. The spatial variables included elevation, slope, peatland area and length, and a topographic position index (TPI). For elevation data, we used the digital elevation model SRTM from NASA, with a 30 m pixel resolution. We obtained the slope as the local gradient computed using the four-connected neighbors of each pixel, expressed in degrees. Additionally, we derived from these elevation data the TPI [47], defined as the difference between the elevation of a pixel and the average elevation of its surrounding pixels.

The spectral variable we estimated was the Normalized Difference Water Index (NDWI). NDWI is a widely used index to assess water content in plants and areas prone to flooding or waterlogging [48–50] and is calculated using the green and near infrared band as $(G-NIR)/(G+NIR)$. For this variable we calculated four descriptive statistics (mean, maximum, minimum and range) using a Landsat 8 surface reflectance collection from January 2019 to December 2021 (30 m pixel resolution). We used NDWI as a proxy to test our predictions concerning peatland water saturation.

## Field surveys

From over 200 peatlands at Vilama, we randomly selected forty peatlands separated by at least 250 m. Selected peatlands were spread over an area of more than 700 km$^2$, located between 4,500 and 4,800 m. a.s.l. and encompass a wide array of sizes (0.4–62 ha) and slopes (3.3–16.8˚) (Fig 1). Peatlands smaller than 0.4 ha were not included in our study. We conducted occupancy surveys between 20 December 2021 and 15 February 2022 to coincide with Diademed Plover's breeding season, when they tend to stay near their nest or flightless chicks, and to satisfy closure assumptions of the occupancy models. As plovers remain active throughout the day, we conducted surveys between 9:00 and 17:00 on days without rain. To assess peatland use, one or two researchers with binoculars walked the entire length of the peatland from bottom (outflow) to top (headwater) and recorded the presence-absence of Diademed Plovers, the number and location of adults, chicks, and nests. Depending on peatland size and location, each survey could last up to 2 hours, and as many as 5–6 peatlands on the same day. We made 2–4 visits to each peatland, separated by at least one day. If we did not find nests we recorded the location of adult plovers on each visit to obtain an approximate location of their territories within peatlands. We used Google Earth to measure the Euclidean distance (horizontal) from the top of the peatland (headwaters) to the nest (if found) or the average location where plovers were recorded. All observations and data collection procedures involving birds were carried out non-invasively and adhered to the principles of animal welfare and conservation [51].

Permits and approvals for fieldwork were obtained from the Ministry of Environment and Climate Change of Jujuy province (Resolución 290/2021-SDBS).

Grazing pressure was estimated from the number of llamas and vicuñas on the peatland at the beginning of each survey. We converted the number of llamas and vicuñas to sheep equivalents to have a common currency for the two species to include in the analyses [52]. Sheep equivalents are based on biomass such that 1 llama = 1.57 sheep equivalents and 1 vicuña = 1.14 sheep equivalents. Thus, our measure of grazing pressure for each peatland was the mean number of sheep equivalents across the 2–4 surveys, divided by peatland area. We detected llamas and vicuñas in the morning and afternoon, and we did not notice a temporal pattern in the use of peatlands by either of these camelids.

## Occupancy modeling

We used single season occupancy models to estimate the proportion of peatlands used by the plovers, and to determine the factors affecting peatland use during the breeding season. Because first time detections by an observer are not independent from subsequent detections by the same observer, we followed an occupancy removal design [53] such that our analysis only included surveys performed until plovers were first detected. Occupancy models estimate the proportion of habitat used when the probability of detection is less than one and have been used extensively to study relatively rare species [31, 54]. Broadly speaking, these are hierarchical models in which occupancy is a binomial state variable and an additional logistic regression is attached to account for imperfect detection [30]. We fit occupancy models using the package SpOccupancy [33] in R version 4.2.3 [55]. The package SpOccupancy fits both spatial and non-spatial occupancy models in a Bayesian framework. Spatial models account for unexplained spatial variation in species occurrence (i.e., unmodeled variation of spatially correlated variables).

Of the 10 peatland characteristics measured (4 descriptive statistics of NDWI, 5 spatial, 1 grazing; see S1 Appendix), we chose four variables to construct our models: peatland area, mean NDWI, grazing, and slope. Slope was the only variable for which we did not have specific predictions and we included it in a more exploratory way. We excluded peatland shape because it was highly correlated with peatland area (r = 0.97). Likewise, topographic index was highly correlated with slope (r = -0.6) and we kept slope as a more intuitive topographic variable. From the spectral variables (NDWI), we used the mean NDWI calculated over three years (2019–2021). NDWI min (the minimum NDWI min recorded over three years) and the range of NDWI (max NDWI – min NDWI) were both correlated with NDWI mean (r = 0.96 and r = -0.57 respectively, S1 Appendix). We decided to use NDWI mean as a more reliable index of the long term moisture dynamics of peatlands. We did not include any isolation metrics as Diademed Plovers are known to move over long distances [26, 28] and therefore plovers could reach even fairly isolated peatlands.

We fit 24 non-spatial models with SpOccupancy and ranked them with the Watanabe Akaike Information criteria (WAIC) (S1 Appendix). Unlike DIC (Deviance Information Criteria), WAIC is an information criteria valid for both hierarchical and mixture models [56, 57] fit in a Bayesian framework. Twelve of the models assumed detection was constant. Each of these twelve models included one or two of the following as covariates of occupancy: peatland area, mean NDWI, grazing, and slope. One model included, as an occupancy covariate, the interaction between peatland area and grazing. The remaining twelve models had the same covariates except that instead of assuming detection was constant, they included peatland area as a detection covariate. All covariates were standardized for statistical modeling using the function scale () in R. We added spatial random effects to the best ranked model in the set of

24 and compared WAIC values between the spatial and non-spatial versions of the model to check if accounting for spatial autocorrelation improved the fit. To estimate random effects in the spatial models we used an exponential covariance function, where UTM coordinates were defined at the centroid of each peatland. For further details on the parametrization of spatial models we made the code and dataset available at github.com/aptrk.

To compute posterior distributions of the parameters we ran three MCMC chains, each with 100,000 iterations, discarded the first half, and used a thinning rate of 20. We used weakly informative priors for all parameters. To check for convergence in our estimates, we visually inspected chain convergence and posterior densities, and ensured that all our parameters showed Gelman-Rubin statistics < 1.1. Finally, to check if our top ranked model fit the data adequately, we conducted posterior predictive checks and estimated Bayesian p-values for both sites and detection histories.

## Results

We detected Diademed Plovers at 18 of the 40 peatlands (naive occupancy 45%), and reproduction events (nests or chicks) at 13 (33% of studied peatlands). We found five nests. When we revisited peatlands where we had previously detected plovers, we always found pairs or lone plovers at the same points where they were detected on previous surveys. Diademed Plovers and nests were more often found closer to the headwaters than to the outflow, except in short peatlands that were located right at the headwater or close to it (Fig 2).

When we modeled peatland occupancy, only two non-spatial models were ranked within 2 WAIC units and both included mean NDWI as a covariate affecting occupancy (Table 1). The top ranked non-spatial model included the effects of slope and mean NDWI as predictors of occupancy, and the effect of peatland area as a predictor of detection. This model was followed closely (0.5 WAIC units) by a simpler model with constant detection and only mean NDWI as

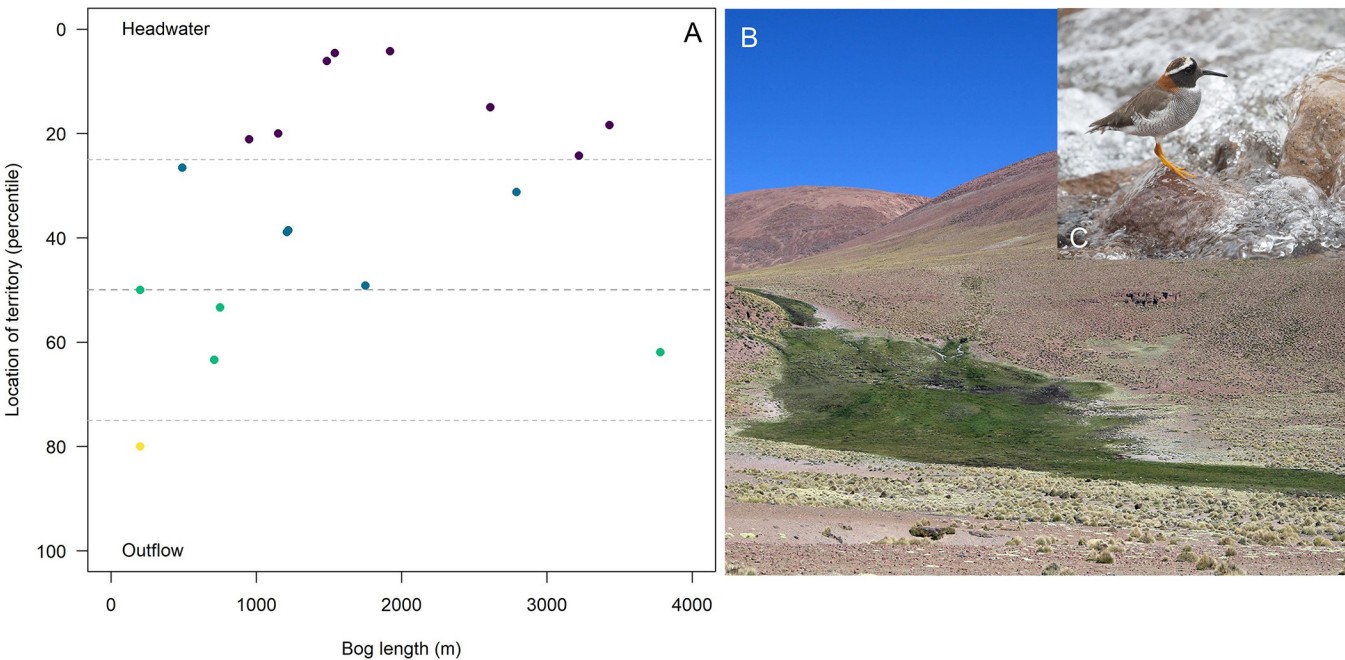

**Fig 2.** A. Location of Diademed Plover's territories as a function of peatland length (measured as the Euclidean distance between the headwaters and the outflow). Dotted horizontal lines indicate 25th, median and 75th percentile. Colors indicate in what percentile territories within these peatlands are located. B. Photograph of a peatland at our study site. C. Diademed Plover (Photos Bruce Lyon).

**Table 1. Estimates and 95% credibility intervals (in parenthesis) of three models predicting occupancy of Diademed Plovers in peatlands of the High Andes.** The table shows results for the two top ranked non-spatial occupancy models and the spatial occupancy model with constant detection and mean NDWI as an occupancy covariate. $\psi$ = occupancy, p = detection. Lower values of WAIC and CV deviance indicate a better fit of the model to the data.

| | Non-spatial | | Spatial |
|---|---|---|---|
| Model | $\psi$(NDWI Mean, Slope) p(Area) | $\psi$(NDWI Mean) p(.) | $\psi$(NDWI Mean) p(.) |
| **Occupancy** | | | |
| Intercept | 0.95 (-0.59, 2.88) | 0.026 (-0.88, 1.34) | 0.321 (-1.05, 1.45) |
| NDWI Mean | -1.96 (-3.77, -0.6) | -1.33 (-2.31, -0.48) | -1.49 (-2.65, -0.55) |
| Slope | 1.19 (-0.38, 3.15) | | |
| **Detection** | | | |
| Intercept | -0.2 (-1, 0.98) | 0.3 (-0.77, 1.36) | 0.31 (-0.74, 1.37) |
| Area | 0.85 (-0.59, 2.13) | | |
| WAIC | 79.3 | 79.8 | 78.6 |
| CV deviance | 81.5 | 83.7 | 83.5 |

a predictor of occupancy. We chose the simpler model for comparison to its spatial version, for two reasons. First, both slope and area showed wide posterior distributions that included 0 within their 95% credible intervals, indicating these parameters were not necessarily informative. Second, a model with fewer parameters was easier to converge with a spatial random effect. The model with spatial random effects ranked slightly better (>1 WAIC units) than its non-spatial equivalent and was also slightly better than the best ranked non-spatial model (Table 1), suggesting some spatial autocorrelation in our dataset, as observed in the S1 Fig under supplementary materials. Similarly, estimates of cross-validation deviance (using a leave one out approach) of the spatial version of the model vs. non-spatial version confirmed a better fit of the spatial version (Table 1). Models that included grazing as a covariate (median grazing: 1.78 sheep equivalents, range: 0–9.79 sheep equivalents) did not perform better than the best ranked model (S1 Appendix). A model that included NDWI mean, and peatland area ranked right after the model that included grazing.

Considering the spatial model, detection of plovers was relatively high (mean: 0.58, CI: 0.32–0.79). Corrected occupancy was 0.51, indicating that more than half of the peatlands are used during the breeding season at Vilama. The spatial model showed an adequate fit to the data, whether the data were binned by site (bayesian p-value 0.69) or by replicate (survey; 0.68). Occupancy of Diademed Plovers increased as mean NDWI values decreased (Table 1, Fig 3). Mean NDWI was highly correlated with minimum NDWI, and detections of Diademed Plovers were also associated with lower minimum NDWI (Fig 3).

## Discussion

Through surveys, remote sensing and occupancy modeling, our study contributes to understanding habitat use and preferences of Diademed Plovers in the High Andes. During the 2021–2022 breeding season, Diademed Plovers occupied about half of the peatlands at Vilama. We also detected plover nests or chicks in 33% of the peatlands surveyed. Plovers did not establish their breeding territories randomly within peatlands, but instead chose locations in the upper portions, near the headwaters. We predicted that occupancy would (1) increase with peatland size and humidity if plovers select larger, water saturated peatlands to maximize foraging resources, and (2) decline with grazing pressure if plovers select ungrazed peatlands to avoid nest trampling by llamas and vicuñas. Occupied peatlands differed in humidity from unoccupied peatlands, supporting the idea that peatlands are not all equivalent to plovers.

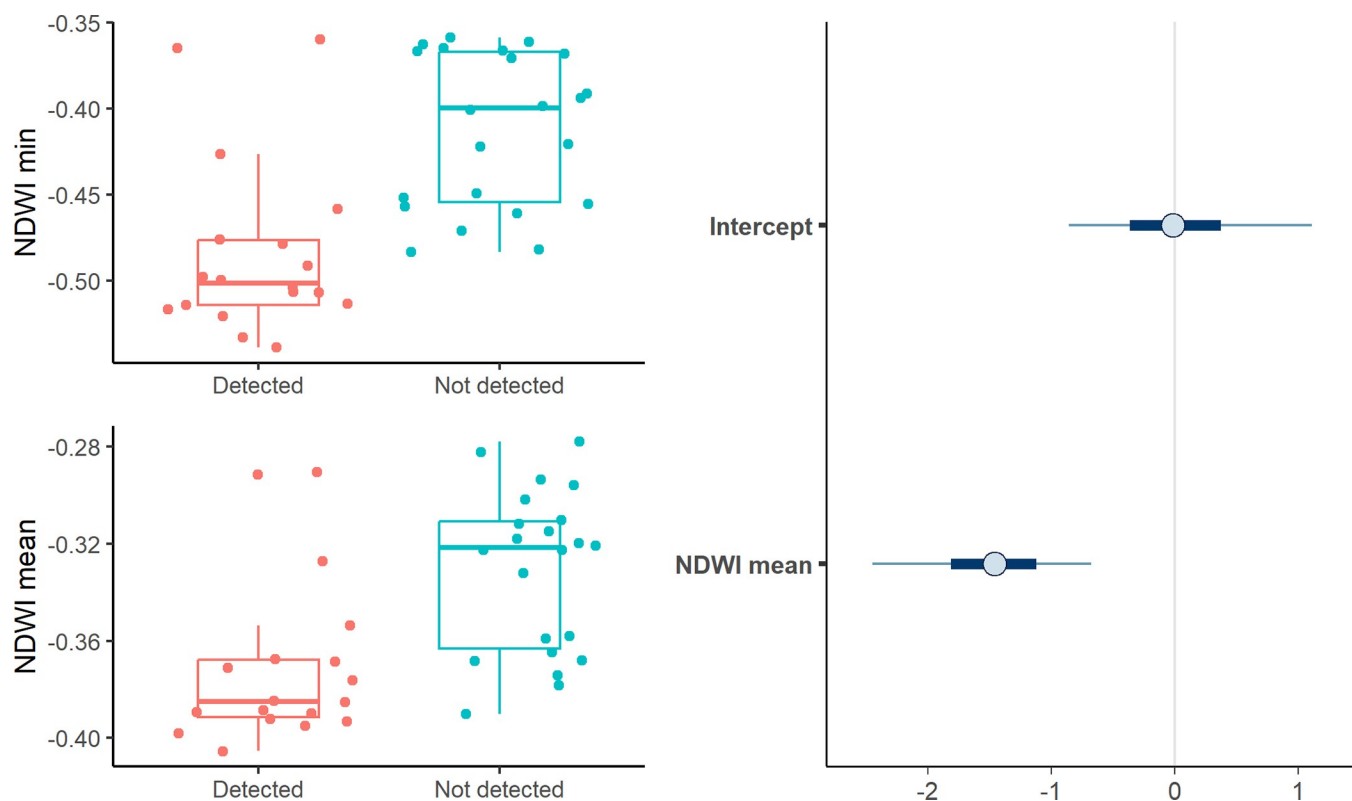

**Fig 3.** Left panel: Raw detection of plovers at peatlands as a function of NDWI min and NDWI mean. The two variables are highly correlated. Right: estimates of the effect of NDWI mean and intercept on occupancy for the spatial model (mean and 90% CI in light blue).

However, contrary to our predictions, Diademed Plovers showed a preference for relatively drier peatlands, and neither peatland size, slope, nor grazing affected their occurrence. Our unexpected findings should stimulate future research on peatland-dwelling organisms of the High Andes and other understudied systems, to expand and test hypotheses about the factors influencing their populations.

Diademed Plovers selected less water-saturated peatlands. This pattern of habitat selection might be a strategy to avoid flooding-induced nest failure, as suggested for other shorebirds [58, 59]. All peatlands where we detected plovers had abundant running water during our surveys, so although used peatlands were relatively drier than unused ones, they were still likely wet enough to have sufficient food availability. Plovers avoided the lower portions of peatlands (outflow, usually near endorheic lakes) and established their breeding territories close to the headwaters. Headwaters tend to be located in steeper terrain compared to the outflow and have reliable water supply as they are close to the water source. Because of the topography of the headwaters, water generally flows in a single, incised stream bed. The lower section of peatlands are comparatively wider, less steep, collect water from other affluents and may be more prone to flooding.

Although Diademed Plovers may occasionally stay year-round at Vilama, the peatlands they use during the breeding season can be frozen for weeks or months outside of the breeding season. Ongoing work [60] suggests that plovers often migrate to lower altitudes during the non-breeding season. At the Vilama wetlands, the lowest NDWI values are always recorded near the end of the dry season (i.e., October), before plovers breed, and precipitation is mostly concentrated in December-March, during the breeding season [18]. By selecting less-saturated

peatlands at the onset of the breeding season, and nesting near the headwaters, Diademed Plovers might reduce their risk of nest flooding as water availability increases in spring. Similarly, Plaschke et al. [59] found that nest initiation of Snowy Plovers (*Charadrius nivosus*) over ten years was associated with lower spring tides (tides that typically occur during full and new moon and show higher amplitude). Nests initiated at those times were more likely to be successful. Flooding has been generally acknowledged as a climatic force affecting the distribution and habitat use of bird species [61]. Extreme events caused by climate change have been shown to increase the risk of nest flooding in several coastal bird species and affected the reproductive output of the Eurasian Oystercatcher (*Haematopus ostralegus*) [62]. Future studies should examine the factors influencing nest survival in Diademed Plovers, with particular attention to variation in flooding risk between and within peatlands.

Contrary to our expectations, neither peatland size nor grazing pressure (by llamas and vicuñas) affected peatland use by Diademed Plovers. Although we predicted that larger peatlands would be more likely to be occupied by plovers, Diademed plovers were often recorded in relatively small peatlands. Diademed Plovers may require only a small foraging area to meet their food requirements. Two observations suggest that Diademed Plovers have small home ranges, which may render peatland size irrelevant to them, above a minimum threshold. First, we consistently observed Diademed Plovers at the same points during our repeated surveys. Second, we observed two different pairs with nests only 250 meters apart on the same peatland. Other plovers of similar size also show relatively small territories that often include several nests per hectare [63–65]. Further studies should examine home range size and minimum peatland size required by Diademed Plovers, including peatlands smaller than the minimum considered in our study (0.4 ha).

Grazing pressure was also absent from the most competitive peatland occupancy models. Two reasons could explain the lack of effects of grazing on plover occupancy. First, llamas are not permanent at Vilama and are present only between late November and early February. Thus, effects of overgrazing and more generally peatland degradation are restricted to only three months per year. Second, densities were overall low (2.84 sheep-equivalents/ha ± 3) and highly variable, as llamas move freely among peatlands (the number of llamas on a peatland varied among surveys at the same peatland). Similar stocking rates of cattle reduced nest survival in Dunlins (*Callidris alpina*) in Finland [37] and Wilson Snipes (*Gallinago gallinago*) [66] in the USA, shorebirds that, like Diademed Plovers, show no defense against cattle but at sites where livestock graze year-round. From a conservation perspective, although we did not examine nest survival, Diademed Plovers have coexisted with short-term Indigenous pastoralism at Vilama for thousands of years, suggesting that even if llamas were a slight threat to nest success, this threat would not be enough to impact the population viability. Considering that the Indigenous pastoral way of life maintains peatland habitat (in the face of mining interests, for example), preserving this way of life may be essential to the goals of plover conservation in the medium term.

Although Diademed Plovers have traditionally been considered very local and rare [28], we suspect their occupancy may have been underestimated. To our knowledge, our study represents the first effort to estimate Diademed Plovers' population state variables (i.e., occupancy). We found surprisingly high occupancy rates; according to the best-ranked model, 50% of the peatlands above 0.4 hectares at Vilama were utilized by Diademed Plovers. Most reported sightings of Diademed Plover (e.g., eBird) are from summer, the period when we observed that plovers use a restricted part of the peatland near the headwaters, which may reduce detections by birdwatchers and other observers. Our finding that adding a spatial random effect improved the fit of the model suggests there are unmodeled spatial or biological variables that affect peatland use by plovers. For example, proximate peatlands tend to exhibit ecological

similarities, influenced by shared soil types, and potentially common water sources, while those irrigated by different mountains may show distinct characteristics shaped by their specific environmental conditions. Alternatively, Diademed Plovers may be more social than thought and may tend to settle in peatlands closer in space. Peatlands similar to the types we studied (type 1 and 2) are widespread in high altitude areas of the central Andes [45], and future studies should look into other peatland types that may also offer habitat for Diademed Plovers. Efforts are underway to assess global populations of Diademed Plovers (Manomet Inc. in preparation). They confirm that the proportion of occupied peatlands at Vilama wetlands is among the highest range-wide, highlighting the importance of Vilama for this globally near-threatened species.

Diademed Plovers selected drier peatlands in the breeding season of 2021–2022, but selection patterns may vary temporally in the face of climate variability that includes El Niño-Southern Oscillation (ENSO) that strongly affects NW Argentina [18, 67]. The 2021–2022 breeding season was among the wettest in the previous decade. In drier years, small peatlands may become unsuitable (no running water) and peatlands that were too humid for Diademed Plovers in 2021–2022 may become suitable, reversing the relationship with NDWI. Given that climate may frequently interact with local factors to affect avian occupancy and demographics [68], it is important to follow up one-season occupancy surveys by replicating occupancy surveys over time, under contrasting conditions, to disentangle the impacts of climate on habitat selection by peatland- and other wetland specialist birds. More generally, understanding interactions between climate and habitat will be critical to understand the effects of climate change on bird populations in the High Andes.

## Supporting information

**S1 Appendix. This appendix describes the correlation among predictors and a table with the 24 non-spatial models ranked by WAIC.**
(DOCX)

**S1 Fig. Vilama wetlands.** Each dot represents a surveyed peatland. Red dots indicate positive spatial random errors, blue dots negative. The size of the dot is proportional to the magnitude of the spatial error.
(TIF)

## Acknowledgments

We thank the Kolla community of Lagunillas del Farallón for allowing us to conduct research on their ancestral territories and the Ministry of Environment and Climate Change of Jujuy province for granting the research permits. Micaela Carral, Facundo Di Sallo, Julián Hernández, Sixto Llampa, Cristian Mamaní, José Segovia and José Sivila helped in the field with the occupancy surveys.

## Author Contributions

**Conceptualization:** Alejandro G. Pietrek, Kristina L. Cockle, Bruce E. Lyon.

**Data curation:** Alejandro G. Pietrek.

**Formal analysis:** Alejandro G. Pietrek, Andrea E. Izquierdo.

**Funding acquisition:** Alejandro G. Pietrek, Kristina L. Cockle, Bruce E. Lyon.

**Investigation:** Alejandro G. Pietrek, Viviana S. Berrios, Bruce E. Lyon.

**Methodology:** Alejandro G. Pietrek, Andrea E. Izquierdo, Bruce E. Lyon.

**Project administration:** Alejandro G. Pietrek.

**Resources:** Alejandro G. Pietrek.

**Supervision:** Alejandro G. Pietrek, Bruce E. Lyon.

**Validation:** Alejandro G. Pietrek.

**Visualization:** Alejandro G. Pietrek.

**Writing – original draft:** Alejandro G. Pietrek, Bruce E. Lyon.

**Writing – review & editing:** Alejandro G. Pietrek, Kristina L. Cockle, Andrea E. Izquierdo, Viviana S. Berrios, Bruce E. Lyon.

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
