## [Decision Letter · Decision Letter 0]

19 Mar 2024

PONE-D-24-00544Habitat occupancy of the threatened Diademed Plover (Phegornis mitchelli) is not affected by llama grazing or peatland size, but declines with peatland humidityPLOS ONE

Dear Dr. Pietrek,

Thank you for submitting your manuscript to PLOS ONE. After careful consideration, we feel that it has merit but does not fully meet PLOS ONE’s publication criteria as it currently stands. Therefore, we invite you to submit a revised version of the manuscript that addresses the points raised during the review process.

We look forward to receiving your revised manuscript.

Kind regards,

Magdalena Ruiz-Rodriguez

Academic Editor

PLOS ONE

Journal Requirements:

2. Thank you for stating the following in the Acknowledgments Section of your manuscript: "This work was funded by a grant of the Neotropical Bird Club and a Maxwell-Hanrahan award to Alejandro Pietrek."

Please remove any funding-related text from the manuscript and let us know how you would like to update your Funding Statement. Currently, your Funding Statement reads as follows: "This work was funded with a grant from the Neotropical Bird Club (https://www.neotropicalbirdclub.org/) awarded to AGP. The sponsors did not play any role in the study design , data collection and analysis, decision to publish or manuscript preparation."

Additional Editor Comments:

Dear Authors,

This paper is suitable for publication, after performing some corrections. The English should be reviewed. Reviewer 1 has made several suggestions to improve the writting, which should be taken in consideration. This reviewer also has concerns about the statistical anañysis, given that it is included a variable (NDVI) that is not related to the predictions. Please, clarify this point.

Reviewer 2 has some concerns about the hypothesis and discussion. I encourage authors to review the manuscript including the suggestions of both reviewers, to improve the manuscript, and then we'll perform a second review.

Yours sincerely

Magdalena Ruiz-Rodríguez

Reviewers' comments:

Reviewer's Responses to Questions

**Comments to the Author**

1. Is the manuscript technically sound, and do the data support the conclusions?

Reviewer #1: Yes

Reviewer #2: Partly

2. Has the statistical analysis been performed appropriately and rigorously? 

Reviewer #1: Yes

Reviewer #2: Yes

3. Have the authors made all data underlying the findings in their manuscript fully available?

Reviewer #1: Yes

Reviewer #2: Yes

4. Is the manuscript presented in an intelligible fashion and written in standard English?

Reviewer #1: Yes

Reviewer #2: No

5. Review Comments to the Author

Reviewer #1: This paper examines the habitat occupancy of a threatened and specialized species, the diademed plover, along peatlands in the Vilamas wetland, in High Andes. The study provides valuable data for the conservation of the diademed plover, especially considering the low information available for this species mainly because of its reluctance to human eye and the difficulty to access to their territories. I do not have major concerns about this paper, the methods are suitable and the main text is clear and concise. However, I have some minor concerns (see below) and some questions about methods. Why using the 4 NDVI variables in the models since you did not have any prediction about this? Indeed, you stated in page 9 that NDVI was included in the models but it was not linked to any initial hypothesis and it did not significantly associated to occupancy. I suggest to either directly remove all methods and analyses in relation to NDVI or to explain why you included NDVI in the models (hypothesis and prediction; in Introduction) and why you finally did not include it in ranked models (in Methods).

Minor comments

Since there is no track-numbering in the manuscript, I numbered text lines for each page. Comments are done following this numbering.

Page 2 – Abstract

Ln 2: Add the scientific name to diademed plover.

Ln 5: “being well-suited to perform occupancy models”.

Ln 6: diademed plovers.

Ln 6: remove may.

Ln 8: diademed plovers.

Ln 12: These results are referred to survey data or model prediction data? If it is referred to survey data, it should be 47%. In this case change to: “Approximately 50% of the studied peatlands…”

Ln 16: Add the meaning of NDWI.

Ln 18: diademed plovers.

Ln 18-20: Maybe join the two results in one sentence, saying that your 2 predictions were not supported by data (namely plover occupancy was positively related to NDWI and negatively related to grazing).

Page 3

Ln 6: “Understanding what factors determine species’ spatial distribution is an important question in population ecology”.

Ln 7: “knowing where a species…”

Ln 12: “Patches may be unoccupied for a number of reasons. For example, due to their abiotic conditions or the presence/absence of other species (1,2)”.

Page 4

Ln 5: “constituting oases within…”

Ln 5: remove “not surprisingly”.

Ln 8: “However, Andean peatlands are now facing rapid changes driven primarily by regional aridization (16-18) and mining activities (19-21)”.

Ln 10: add the species name of the invasive trout.

Ln 11: “To address the consequences of these threats, we need…”

Ln 17: 4,000 m above sea level (m a.s.l.).

Page 5

Ln 5: remove “if any”.

Ln 13: add coma after random effects.

Ln 17: “are hypothetised to harbour…”

Page 6

Ln 3: diademed plover.

Ln 4: add the scientific name of vicuñas (Lama vicugna). Also, scientific name of llama is incorrectly written (Lama glama).

Ln 7: “The aim of this paper…”

Ln 14: “to ask whether they…”

Ln 17: plovers instead of birds.

Page 7

Ln 5: Do not mention here that you performed surveys, since they are described later in methods. Instead, you should start describing the study are (e.g. The study region is located at Lagunas de Villama…).

Ln 9: “It was declared Ramsar site in 2000, and Important Bird Area (IBA) in … (add year)”.

Ln 12: Maybe “arid cold desert”.

Ln 12: “an extreme temperature amplitude”.

Ln 14: “ranging from 100 to 300 mm per year. There are often…”

Page 8

Ln 3: Maybe explain briefly types 1 and 2 to understand better the peatland ecosystem.

Ln 5: Also, maybe explain briefly the Kolla (or Qulla) indigenous community; i.e. from where they come, their transhumance, etc. In this sense, non-native people could understand better the Andean peatland system.

Figure 1: Add the scale in the two maps and the 4 cardinal directions.

Ln 10: “and females lay 2 eggs…”

Ln 13: provision or feed instead of raise.

Ln 15: plovers instead of they.

Page 9

Ln 4: “using the Google Earth Engine (GEE) platform”.

Ln 6: topographic position index (TPI).

Ln 9: use only TPI, as you described it before.

Ln 11: “The spectral variables include…” Also, remove all methods concerning NDVI (see my general comments).

Ln 13: add the resolution in pixels of Lansat-8 images.

Ln 14: explain how NDWI is calculated from Landast-8 bands. NDWI = (B3-B4)/(B3+B4).

Ln 15: “We used NDWI as proxy to test our predictions…”

Page 10

Ln 9: researchers instead of people.

Ln 14: add the ID of permits/approval (here or in an ethic statement at the end of the manuscript).

Ln 15: “The grazing pressure was estimated as the number of…”

Ln 16: Did you note the presence of llamas/vicuñas in the afternoon while doing surveys? You mention that surveys were done until 17:00h. If no llama/vicuña was detected along the day, you should mention it.

Ln 17: better not to refer to sheep equivalents as SE (because it can be confused with standard error). Simply refer as sheep equivalents.

Page 11

Ln 6: add coma after observer.

Ln 12: add the R version

Ln 15: Number the 14 variables (8 statistics NDWI, NDVI, 5 spatial, grazing).

Ln 16: Why is the reasoning behind the selection of the 4 variables selected?

Page 12

Ln 5: reference the Appendix 1 when you mention the 24 fitted models.

Ln 8: This sentence is a bit confusing, please, re-write.

Ln 11: How covariates were standardized? Scaling? Describe it briefly.

Page 13

General comment in results: why you do not present the data relative to grazing of llamas and vicuñas (sheep-equivalents)? I suggest to add descriptive statistics in results, as you mention later in the discussion some of the results.

Ln 4: diademed plovers.

Ln 4: 18 out of 40.

Ln 5: 13 out of 40.

Ln 6-9: move this part to methods and report here only the results relative to figure 2.

Ln 15: Photograph instead of view. Also, it would be better if the margins of the plover photo match with the margins of the peatland photo.

Ln 18: “both included NDWI as a covariate affecting occupancy”.

Page 14

Ln 8: “and was also slightly better than the best ranked model (Table 1)”.

Ln 11-14: You can remove this part and shorten saying that models including grazing as covariate did not perform better than the best ranked model (see appendix 1).

Page 15

Table 1: Add a new row in the top to separate the two non-spatial models and the spatial model.

Ln 1: “95% confidence intervals (in parentheses) of three models…”

Ln 2: “The table shows results…”

Ln 6: indicate that 0.58 is the estimate and 0.32-0.79 is the 95%-CI.

Ln 7: indicating instead of suggesting.

Ln 10: remove “i.e. humidity”. Also change to: “(Table 1; Fig. 3)”.

Page 16

Figure 3: Better refer to Fig3a and Fig3b instead of left and right panel. Also remove the correlation in the figure legends as it is posted in the main text.

Line 10: 33% instead of 32% (to report the same % as in results).

Ln 11: “chose locations in their upper portions, near the headwaters”.

Ln 12: number the two predictions: “We predicted that: (1) occupancy would increase…; and (2) occupancy would decline…”

Ln 15: “unoccupied peatlands in their abiotic conditions, supporting…”

Ln 16: two predictions instead of all predictions.

Page 17

Ln 4-8: In this part of the discussion, you state that drier peatlands should support a lower food availability for plovers than more humid peatlands. However, neither in the introduction nor here you show a clear relationship between these two variables. The NDWI is a surrogate of humidity, but maybe it is not related to food abundance and/or availability in your study system. I would add that more studies are needed to examine the food resources for plovers in weatlands differing in humidity (or NDWI), so your results are maybe not “unexpected”.

Ln 19: separate into two sentences.

Page 18

Ln 1: Your ongoing work (58) examines the within-peatland migration or altitudinal migration in the Andes? If the first option is correct, I understand why plovers choose headwaters instead of lower parts of peatlands.

I think references 58 and 59 are incorrectly numbered (58 should be 59 and viceversa).

Ln 4: add scientific name (Charadrius nivosus).

Ln 5: “with nests being initiated at those times…”

Ln 7: start a new sentence after the reference 60.

Ln 8: “nest flooding in several coastal species (61)”.

Ln 12: should examine.

Ln 16: separate into two sentences: “food availability. Diademed plovers…”

Ln 17: Two observations would support this hypothesis”.

Page 19

Ln 1: “include” is duplicated.

Ln 3: should examine. Also, start this new paragraph directly with the grazing.

Ln 5: Two reasons could explain the lack…”

Ln 5: I do not see how the first reasoning can invalidate your findings, since llamas forage in the same period as plovers reproduce (December-February). One could expect some competitive exclusion by llamas, as they would pressure plover nests and/or territories.

Ln 8: sheep-equivalents instead of se.

Ln 8: in relation to the two reasonings, the lack of effect could be attributable to the fact that llamas and vicuñas were counted only in the morning, when starting the surveys. Maybe if you surveyed them during all day, grazing could affect plover occupancy. See my comment above.

Ln 10: add the scientific names of dunlins and snipes.

Ln 13: even if llamas exert a slight threat to nest success, this…”

Ln 15: “pastoral way of life (…), may be essential for Diademed plover conservation in the mid-term”.

Ln 20: remove the colon.

Page 20

Ln 11: “Vilama peatlands are among the highest proportion of occupied peatlands across the species range”.

Ln 15: “climate variability, including the El Niño-Southern Oscillation (ENSO) which strongly affects NW Argentina”.

Ln 19: “affect bird’s occupancy and demography”.

Reviewer #2: The goal of this paper is to identify peatland features that Diademed Plovers use during their breeding season when they establish delimited territories within peatlands. This study is intriguing, particularly for a species with limited information on its biology and ecology. I commend the authors for the extensive sampling effort undertaken for this research. I believe that the manuscript is publishable, but from my point of view, it could benefit from restructuring to organize ideas and enhance readability. Also, I am not a native English speaker, but I believe it requires an English formatting review.

Some examples I suggest are:

Introduction:

1- Rewrite the hypotheses paragraph, being more concise: "The hypotheses in this work are… The predictions are…"

2- Are there no other predation pressures on these sites for plovers aside from grazing?

3- On page 5, the sentence: “Importantly, their specific micro-habitat requirements remain largely unknown, and Diademed Plovers are often absent from peatlands that, to the human eye, seem identical to those where individuals are present (28)” It is not clear what is meant.

4- I recommend omitting the word “rare” when referring to the species, as it may confuse due to its conservation status, in the introduction section.

5- The species could be further characterized, by detailing their wintering and breeding areas. It would be beneficial to specify at which stage of the life cycle they inhabit these areas. Exploring their cohabitation with other species, potential competition, interrelations with other birds and/or animals, as well as the impact of grazing, would also be valuable.

Methods:

1- Sentence: “Peatlands at Vilama can be classified as types 1 and 2 (44): they are generally dominated by plant communities adapted to high altitudes and extreme conditions and located in relatively steep areas” What would be types 1 and 2?

2- Sentence: “Each summer between late November and early February, Kolla shepherds from the neighboring community of Lagunillas del Farallón move up to Vilama to graze their llamas on the peatlands, but no people live at Vilama year- round.” I assume these are the same months plovers occupy the breeding area, but that should be clear.

3- Sentence: “Diademed Plovers make a rudimentary nest on the ground and lay two eggs in late spring. Diademed Plovers both forage and nest within the stream habitats; nests are placed on the cushion plants or among rocks in rocky patches. Both parents incubate eggs and raise the chicks, which are precocial. The breeding grounds of Diademed Plovers are often inaccessible to observers during the winter, and winter distribution data are scarce. In areas like Vilama, where many peatlands freeze in winter, they might be altitudinal migrants.” This information might be better moved to the introduction for a clearer understanding of the species' ecology.

Discussion:

1- Sentence: “During the 2021–2022 breeding season, Diademed Plovers occupied about half of the peatlands at Vilama”. This would be half of the peatlands selected by the authors, right?

2- Sentence: “Although we predicted that peatland size would increase food availability to Diademed Plovers, we did not measure food availability, and Diademed Plovers may require only a small foraging area to meet their food requirements”. I recommend refraining from making predictions in this case, as without analyzing food availability, predicting outcomes in a changing environment is not feasible. While the plover may need relatively small foraging areas, it is possible that during the sampling period, prey availability was insufficient, or specific prey types or sizes were lacking. Without trophic studies, it may be best to avoid speculating on hypotheses, particularly for a species with limited knowledge and difficult access to research. It would be beneficial to note observations for future studies, without proposing specific hypotheses.

3- Sentence: “Grazing pressure was also absent from the most competitive peatland occupancy models. Two reasons stand out to explain the lack of effects of grazing on plover occupancy. First, llamas are not permanent at Vilama and are present only between late November and early February. Second, densities were overall low (2.84 se/ha ± 3) and highly variable, as llamas move freely among peatlands (the number of llamas on a peatland varied among surveys at the same peatland)”. However, they are present and moving randomly precisely during the breeding months, leading the authors to formulate a hypothesis with their prediction. Perhaps this occurrence was coincidental, or the llamas are in wetter peatlands, or the number of llamas grazing is insignificant for the plovers.

This is the proposal put forward by the authors later: “From a conservation perspective, although we did not examine nest survival, Diademed Plovers have coexisted with short-term Indigenous pastoralism at Vilama for thousands of years, suggesting that even if were llamas are a slight threat to nest success, this threat would not be enough to impact the population viability. Moreover, preserving the Indigenous pastoral way of life (in the face of mining interests, for example), and the resultant habitat preservation this maintains, may be essential to the goals of plover conservation in the medium term. Diademed Plovers selected drier peatlands in the breeding season of 2021–2022, but selection patterns may vary temporally in the face of climate variability that includes the ENSO that strongly affects NW Argentina (18,66). The 2021–2022 breeding season was among the wettest in the previous decade. In drier years, small peatlands may become unsuitable (no running water) and peatlands that were too humid for Diademed Plovers in 2021–2022 (large NDWI values) may become suitable. Given that climate may frequently interact with local factors to affect occupancy and demographics of birds (67), it is important to follow up one-season occupancy surveys by replicating occupancy surveys over time, under contrasting conditions, to disentangle the impacts of climate on habitat selection by peatland- and other wetland specialist birds. More generally, understanding interactions between climate and habitat will be critical to understand the effects of climate change on bird populations in the High Andes.”

This is crucial. I recommend beginning with these statements to form hypotheses and predictions, as this data does not arise from this study; instead, it should serve as the foundation of this study, just like the previous data on grazing.

6. PLOS authors have the option to publish the peer review history of their article (what does this mean?). If published, this will include your full peer review and any attached files.

Reviewer #1: No

Reviewer #2: No

---

## [Author Response · Author response to Decision Letter 0]

17 Apr 2024

#Reviewer 1

This paper examines the habitat occupancy of a threatened and specialized species, the diademed plover, along peatlands in the Vilamas wetland, in High Andes. The study provides valuable data for the conservation of the diademed plover, especially considering the low information available for this species mainly because of its reluctance to human eye and the difficulty to access to their territories. I do not have major concerns about this paper, the methods are suitable and the main text is clear and concise. However, I have some minor concerns (see below) and some questions about methods. Why using the 4 NDVI variables in the models since you did not have any prediction about this? Indeed, you stated in page 9 that NDVI was included in the models but it was not linked to any initial hypothesis and it did not significantly associated to occupancy. I suggest to either directly remove all methods and analyses in relation to NDVI or to explain why you included NDVI in the models (hypothesis and prediction; in Introduction) and why you finally did not include it in ranked models (in Methods).

We have removed references to NDVI throughout the text. We did not incorporate NDVI among the variables of the occupancy models. We did visually explore Diademed Plover occurrence as a function of NDVI descriptive statistics, but we have removed this from the paper to focus only on the variables that we did include in occupancy models.

Minor comments

Since there is no track-numbering in the manuscript, I numbered text lines for each page. Comments are done following this numbering. 

Page 2 – Abstract

Ln 2: Add the scientific name to diademed plover.

R: We added the scientific name. (Line 4)

Ln 5: “being well-suited to perform occupancy models”.

R: We changed the sentence to read "Andean peatlands are well-suited to occupancy modelling because they are discrete patches of humid habitat within a matrix of high-altitude steppe”. Lines (Lines 5-7)

Ln 6: diademed plovers.

R: We rewrote this as Diademed Plovers. (Line 7)

Ln 6: remove may.

R: We removed “may”. (Line 7)

Ln 8: diademed plovers.

R: We rewrote this as Diademed Plovers. (Line 9)

Ln 12: These results are referred to survey data or model prediction data? If it is referred to survey data, it should be 47%. In this case change to: “Approximately 50% of the studied peatlands…”

R: We changed the sentence to read “Occupancy models showed that more than 50 % of the studied peatlands were used by Diademed Plovers”, to clarify that the statement refers to model predictions. (Lines 14-15).

Ln 16: Add the meaning of NDWI.

R: We clarified the meaning of NDWI: "Normalized Difference Water Index, an index correlated with water content and humidity". (Lines 18-19)

Ln 18: diademed plovers.

R: Changed to Diademed Plovers. (Line 20)

Ln 18-20: Maybe join the two results in one sentence, saying that your 2 predictions were not supported by data (namely plover occupancy was positively related to NDWI and negatively related to grazing).

R: No change made. We considered the suggestion but found the text clearer when we kept these as two concise and shorter sentences.

Page 3

Ln 6: “Understanding what factors determine species’ spatial distribution is an important question in population ecology”.

R: We changed the sentence to read " A major aim of population ecology is to identify the factors that determine how species are distributed in space”. (Lines 32-33)

Ln 7: “knowing where a species…”

R: We replaced “understanding” with “knowing” following the reviewer’s advice. (Line 33)

Ln 12: “Patches may be unoccupied for a number of reasons. For example, due to their abiotic conditions or the presence/absence of other species (1,2)”.

R: We changed the structure of the paragraph to indicate the three main reasons: First,.... Second,..... Third… (Line 38)

Page 4

Ln 5: “constituting oases within…”

R: We changed the section to read: " Peatlands in the High Andes of South America occur as discrete, humid habitat patches along streams within an arid matrix of high-altitude steppe. These peatlands are oases that harbor highly distinct biological communities, including many endemic amphibians and birds (10–12)."(Lines 48-49)

Ln 5: remove “not surprisingly”.

R: We removed "not surprisingly."(Line 48)

Ln 8: “However, Andean peatlands are now facing rapid changes driven primarily by regional aridization (16-18) and mining activities (19-21)”.

R: We changed the sentence to read: " However, these biodiversity islands are experiencing rapid changes in vegetation related to declining humidity, caused by aridization of the climate at a regional scale (16–18) and escalating water requirements for mining activities at local scales (19–21). "(Lines 50-53)

Ln 10: add the species name of the invasive trout.

R: Added. (Line 54)

Ln 11: “To address the consequences of these threats, we need…”

R: We have now re-written the sentence following the reviewer suggestion. (Line 55)

Ln 17: 4,000 m above sea level (m a.s.l.).

R: Added (Line 61)

Page 5

Ln 5: remove “if any”.

R: We removed "if any". (Line 71)

Ln 13: add coma after random effects.

R: Added. (Line 80)

Ln 17: “are hypothetised to harbour…”

R: No change made. We are linking the formal theory and evidence on patch size/population size to peatlands and describing where our system might fit. There are no formal hypotheses on peatland size effects on occupancy. (Line 86)

Page 6

Ln 3: diademed plover.

R: We rewrote as Diademed Plovers. (Line 108)

Ln 4: add the scientific name of vicuñas (Lama vicugna). Also, scientific name of llama is incorrectly written (Lama glama).

R: We have corrected the scientific name of lamas and added the scientific name for vicuñas. According to the red list published by the Argentinean Society of Mammalogists the scientific name of vicuñas is still Vicugna vicugna.(Arzamendia, Y.; Acebes, P.; Baldo, J. .; Rojo, V. ; Segovia, J.M. 2019) (Lines 93-96)

Ln 7: “The aim of this paper…”

R: We rewrote the sentence following the suggestion of the reviewer. (line 101)

Ln 14: “to ask whether they…”

R: We changed the sentence to read " Additionally, we explored the locations, within peatlands, used by the Diademed Plovers, to determine whether plovers were distributed randomly or toward one end of these somewhat linear habitats." (Lines 107-109)

Ln 17: plovers instead of birds.

R: We replaced “birds” with “plovers”. (Line 111)

Page 7

Ln 5: Do not mention here that you performed surveys, since they are described later in methods. Instead, you should start describing the study are (e.g. The study region is located at Lagunas de Villama…).

R: We have now changed the sentence and it now reads “We conducted our study at…” (Line 120)

Ln 9: “It was declared Ramsar site in 2000, and Important Bird Area (IBA) in … (add year)”. 

R: Added (Line 124)

Ln 12: Maybe “arid cold desert”.

R: We have followed here the description of Peel et al. 2007 that classifies the climate type as “arid desert cold”.

Ln 12: “an extreme temperature amplitude”.

R: Done. (Line 127)

Ln 14: “ranging from 100 to 300 mm per year. There are often…”

R: We have now split this in two sentences as suggested by the reviewer. (Line 129)

Page 8

Ln 3: Maybe explain briefly types 1 and 2 to understand better the peatland ecosystem.

R: We have now specified the two cushion plants that characterize these peatlands (Distichia and Oxychloe), and further detail they occur in relatively steep areas above 4,000 m . (Lines 136-138)

Ln 5: Also, maybe explain briefly the Kolla (or Qulla) indigenous community; i.e. from where they come, their transhumance, etc. In this sense, non-native people could understand better the Andean peatland system. 

R: We have now added one sentence to describe general details on the Kollas people. The sentence now reads: “Each summer between late November and early February, Kolla shepherds from the neighboring community of Lagunillas del Farallón move up to Vilama to graze their llamas on the peatlands, but no people live at Vilama year-round. The Kollas comprise a group of tribes that inhabit northwest Argentina, northern Chile, and Bolivia and have held common land property in this area for centuries”. (Lines 141-143)

Figure 1: Add the scale in the two maps and the 4 cardinal directions.

R: We added the scale and north arrow.

 Ln 10: “and females lay 2 eggs…”

R: We added "females" to the original sentence. (Lines 64-65)

Ln 13: provision or feed instead of raise.

R: We changed it to 'provision'. (Line 66)

Ln 15: plovers instead of they.

R: We replaced “they” with “plovers” (Line 68)

Page 9

Ln 4: “using the Google Earth Engine (GEE) platform”.

R: Corrected. (Line 155)

Ln 6: topographic position index (TPI).

R: Corrected. (Line 156)

Ln 9: use only TPI, as you described it before.

R: Corrected. (Line 159)

Ln 11: “The spectral variables include…” Also, remove all methods concerning NDVI (see my general comments).

R: We removed the sentences about NDVI throughout the text because we did not incorporate NDVI statistics in the occupancy models. 

Ln 13: add the resolution in pixels of Lansat-8 images.

R: We added the resolution of the Landsat imagery. (Line 165)

Ln 14: explain how NDWI is calculated from Landast-8 bands. NDWI = (B3-B4)/(B3+B4).

R: We have now added the NDWI equation in the methods (Lines 158-160 of the revised version). The sentence now reads: “NDWI is a widely used index to assess water content in plants and areas prone to flooding or waterlogging (47–49) and is calculated using the green and near infrared band as (G-NIR)/(G+NIR)”. (Lines 161-163)

Ln 15: “We used NDWI as proxy to test our predictions…”

R: Corrected. (Line 166)

Page 10

Ln 9: researchers instead of people.

R: We now use researchers instead of people. (Line 177)

Ln 14: add the ID of permits/approval (here or in an ethic statement at the end of the manuscript).

R: We have now added the permit numbers here and the ethics statement. (Lines 185-188)

Ln 15: “The grazing pressure was estimated as the number of…”

R: We changed the phrase to "Grazing pressure was estimated from the number of..." (Line 189)

Ln 16: Did you note the presence of llamas/vicuñas in the afternoon while doing surveys? You mention that surveys were done until 17:00h. If no llama/vicuña was detected along the day, you should mention it.

R: We added the following sentence: "We detected both llamas and vicuñas in the morning and afternoon, and we did not notice a temporal pattern in the use of peatlands by these camelids." (Lines 194-195)

Ln 17: better not to refer to sheep equivalents as SE (because it can be confused with standard error). Simply refer as sheep equivalents.

R: We changed to "sheep equivalents" throughout the manuscript.

Page 11

Ln 6: add coma after observer.

R: Added. (Line 201)

Ln 12: add the R version

R: Added. (Line 207)

Ln 15: Number the 14 variables (8 statistics NDWI, NDVI, 5 spatial, grazing).

R: We clarified what the ten (after removing NDVI statistics) variables are. (Line 211)

Ln 16: Why is the reasoning behind the selection of the 4 variables selected?

R: We have justified the use of area and grazing in the introduction. However we agree that the use of mean NDWI as the chosesn statistic merits further explanation. In lines 216-217 we wrote: “NDWI min (the minimum NDWI min recorded over three years and the range of NDWI ( max NDWI- min NDWI) were both correlated with NDWI mean, but we decided to use NDWI mean as a more reliable statistics of the long term moisture dynamics of peatlands.” (Lines 216-218) 

As for slope in lines 213-214 we wrote: “Slope was the only variable for which we did not have specific predictions and we included it in a more exploratory way”.

Page 12

Ln 5: reference the Appendix 1 when you mention the 24 fitted models.

R: Reference added. (Line 223)

Ln 8: This sentence is a bit confusing, please, re-write.

R: We rewrote the section as follows: "Twelve of the models assumed detection was constant. Each of these twelve models included one or two of the following as covariates of occupancy: peatland area, mean NDWI, grazing, and slope. One model included, as an occupancy covariate, the interaction between peatland area and grazing. The remaining twelve models were basically the same except that instead of assuming detection was constant, they included peatland area as a detection covariate”. (Lines 225-229)

Ln 11: How covariates were standardized? Scaling? Describe it briefly.

R: The sentence now reads: “All covariates were standardized for statistical modeling using the function scale () in R.” (Line 229-230)

Page 13

General comment in results: why you do not present the data relative to grazing of llamas and vicuñas (sheep-equivalents)? I suggest to add descriptive statistics in results, as you mention later in the discussion some of the results.

R: We added median values and range of the Grazing variable. (Lines 266-267)

Ln 4: diademed plovers.

R: Added. (Line 242)

Ln 4: 18 out of 40.

R: No change made. "18 of 40" is grammatically correct and more concise. (Line 242)

Ln 5: 13 out of 40.

R: No change made. "13 of 40" is also correct and more concise. (Line 243)

Ln 6-9: move this part to methods and report here only the results relative to figure 2.

R: We moved the sentence " We used Google Earth to measure the Euclidean distance (horizontal) from the top of the peatland (headwaters) to the nest (if found) or the average location where plovers were recorded" to lines 183-184 under methods.

Ln 15: Photograph instead of view. Also, it would be better if the margins of the plover photo match with the margins of the peatland photo.

R: We changed “View” to “Photograph”. The margins of the plover photo now fit the peatland photo (Line 251)

Ln 18: “both included NDWI as a covariate affecting occupancy”. 

R: Changed. (Line 254)

Page 14

Ln 8: “and was also slightly better than the best ranked model (Table 1)”. 

R: We changed the phrase to "and was also slightly better than the best ranked model (Table 1)". (Line 261)

Ln 11-14: You can remove this part and shorten saying that models including grazing as covariate did not perform better than the best ranked model (see appendix 1).

R: We rewrote the sentence: "Models that included grazing as a covariate (median grazing: 1.78 sheep equivalents, range: 0-9.79 sheep equivalents) did not perform better than the best ranked non-spatial model (Appendix I).". (Lines 265-267)

Page 15

Table 1: Add a new row in the top to separate the two non-spatial models and the spatial model.

R: Added. (Table 1)

Ln 1: “95% confidence intervals (in parentheses) of three models…”

R: Added. (Caption Table 1)

Ln 2: “The table shows results…”

R: Changed. (Caption Table 1)

Ln 6: indicate that 0.58 is the estimate and 0.32-0.79 is the 95%-CI.

R: Added. (Line 272)

Ln 7: indicating instead of suggesting.

R: Changed. (Line 272)

Ln 10: remove “i.e. humidity”. Also change to: “(Table 1; Fig. 3)”.

R: Removed. (Line 276)

Page 16

Figure 3: Better refer to Fig3a and Fig3b instead of left and right panel. Also remove the correlation in the figure legends as it is posted in the main text.

R: Correlation removed from the figure legend. As the left panel comprises two plots, we prefer to leave the reference as in the original version. (Caption Figure 3)

Line 10: 33% instead of 32% (to report the % as in results).

R: Corrected. Thank you for the observation. (Line 286)

Ln 11: “chose locations in their upper portions, near the headwaters”.

R: We rewrote the sentence: "Plovers did not establish their breeding territories randomly within peatlands, but instead chose locations in the upper portions, near the headwaters." (Lines 286-287)

Ln 12: number the two predictions: “We predicted that: (1) occupancy would increase…; and (2) occupancy would decline…”

R: Added. (Lines 287-291)

Ln 15: “unoccupied peatlands in their abiotic conditions, supporting…”

R: We changed the sentence to read " Occupied peatlands differed in humidity from unoccupied peatlands". (Line 291)

Ln 16: two predictions instead of all predictions.

R: We rewrote these as “contrary to our predictions”. Within the sentence, we more 

---

## [Decision Letter · Decision Letter 1]

15 May 2024

PONE-D-24-00544R1Habitat occupancy of the threatened Diademed Plover (Phegornis mitchellii) is not affected by llama grazing or peatland size, but declines with peatland humidityPLOS ONE

Dear Dr. Pietrek,

Thank you for submitting your manuscript to PLOS ONE. After careful consideration, we feel that it has merit but does not fully meet PLOS ONE’s publication criteria as it currently stands. Therefore, we invite you to submit a revised version of the manuscript that addresses the points raised during the review process.

We look forward to receiving your revised manuscript.

Kind regards,

Vitor Hugo Rodrigues Paiva, Ph.D.

Academic Editor

PLOS ONE

Journal Requirements:

Reviewers' comments:

Reviewer's Responses to Questions

**Comments to the Author**

1. If the authors have adequately addressed your comments raised in a previous round of review and you feel that this manuscript is now acceptable for publication, you may indicate that here to bypass the “Comments to the Author” section, enter your conflict of interest statement in the “Confidential to Editor” section, and submit your "Accept" recommendation.

Reviewer #1: All comments have been addressed

Reviewer #3: (No Response)

Reviewer #4: (No Response)

2. Is the manuscript technically sound, and do the data support the conclusions?

Reviewer #1: Yes

Reviewer #3: Yes

Reviewer #4: Yes

3. Has the statistical analysis been performed appropriately and rigorously? 

Reviewer #1: Yes

Reviewer #3: Yes

Reviewer #4: Yes

4. Have the authors made all data underlying the findings in their manuscript fully available?

Reviewer #1: Yes

Reviewer #3: Yes

Reviewer #4: Yes

5. Is the manuscript presented in an intelligible fashion and written in standard English?

Reviewer #1: Yes

Reviewer #3: Yes

Reviewer #4: Yes

6. Review Comments to the Author

**Reviewer #1**: The authors correctly addressed the review. I do not have additional comments, since the manuscript is now acceptable for publication.

**Reviewer #3**: This revised paper provides valuable ecological information for an understudied species in a unique system. The study design and analysis are robust, and the writing is clear. I only have minor suggestions below.

Line 16: Are there other plover species that occupy Vilama and the peatlands are important for their conservation as well? If you are referring to diademed plovers, perhaps change ‘plover conservation’ to ‘the conservation of this species’.

Line 20: I do think the last paragraph of the Discussion provides some needed context that the 2021-2022 breeding season was wetter than average years and this could be noted in the Abstract. It felt like a big reveal at the very end of the paper that would have been nice to know up front.

Line 87: Perhaps ‘greater diversity of micro-habitats’ instead of ‘more’

Figure 1: Maybe it’s just the resolution of the image that I am seeing in the document, but the different dot sizes are not very apparent and I’m wondering if size is worth including in the Figure without a legend indicating the magnitude of variation. In the caption, does the color refer to the latent spatial random effect (meaning more plovers in the red dots and less plovers in the blue dots)? I am not sure how to interpret “spatial random error”, but I would suspect you mean to say the value of the spatial effect at each site (positive or negative). It’s also interesting that there seems to be an east to west effect where there were more plovers in the eastern sites and less in the western sites. I’m wondering if the authors can hypothesize why there seems to be some clustering almost split in the middle of the surveyed sites.

Line 126-130: Related to my previous comment, it would be beneficial to add how precipitation (and maybe temperature) during the study year compared to these general descriptions of climate conditions.

Line 228: Rather than ‘basically the same’, state that the covariates on occupancy were the same as the 12 models with constant detection probabilities.

Line 230: What coordinates were used for the spatial random effect? Was it the centroid of each peatland, the starting point, or something else? The centroid would seem to be the least affected by the size of the peatland.

Line 234: Change ‘make’ to ‘made’

Line 260: Easier to fit in what way?

**Reviewer #4: **This paper presents new data on factors that affect peatland use by a little-studied species, the Diademed Plover. The study is overall well-executed, and the statistical analyses are appropriate for the study system and research questions. I do not have any major concerns, and the authors have done a commendable job addressing previous reviewer questions. My only general question is to ask the authors how variable was grazing pressure among each peatland? Were there some peatlands that, for example, experienced zero grazing pressure, while others experienced heavy grazing pressure? And are counts of vicunas and llamas during each survey sufficient to characterize the grazing pressure experienced throughout the course of the season? I wonder if an effect could be detected if grazing pressure were measured using different methods, or if there were a wider range of grazing pressures present within the study system.

Minor editorial comments given below:

Abstract

Lines 7-10: These two sentences are somewhat repetitive and can be condensed.

Introduction

Lines 74-76: This sentence is somewhat awkward and should be restructured.

Paragraph lines 74-82: This paragraph provides some general discussion of modeling occurrence/occupancy. It could use another sentence or two at the end to explain the relevance of this background information to the current study, perhaps by explaining the appropriateness of the study system for this type of occupancy model.

Line 95: coastal habitats?

Lines 105-107: These two sentences are somewhat repetitive and can be condensed.

Methods

Line 137: What are these types?

Line 155: included

Line 172: encompassing

Results

Lines 276-278: The first half of this sentence could go to Methods section, paragraph lines 211-221 where this correlation is discussed. Also, I’m not sure why the plot of detections vs. NDWI min is included in Figure 3, given that this variable was discarded from the occupancy models due to correlation with NDWI mean.

Discussion

Line 315: Snowy Plovers

Line 320: Oystercatcher

Line 325: Plovers

Line 330: similarly-sized

Line 340: Dunlins and Wilson’s Snipes

Line 341: “…shorebirds that show no defense against cattle like the Diademed Plovers” – wording is somewhat unclear. I suggest either omitting, or rewording as “shorebirds that, like the Diademed Plovers, show do defense…”

Lines 369-371: Which could lead to a positive association with NDWI in drier years – perhaps important to note.

7. PLOS authors have the option to publish the peer review history of their article (what does this mean?). If published, this will include your full peer review and any attached files.

Reviewer #1: **Yes: **Jorge Garrido Bautista

Reviewer #3: No

Reviewer #4: No

---

## [Author Response · Author response to Decision Letter 1]

28 May 2024

Reviewer 3: This revised paper provides valuable ecological information for an understudied species in a unique system. The study design and analysis are robust, and the writing is clear. I only have minor suggestions below.

Line 16: Are there other plover species that occupy Vilama and the peatlands are important for their conservation as well? If you are referring to diademed plovers, perhaps change ‘plover conservation’ to ‘the conservation of this species’.

R: We changed it to “Diademed Plover conservation” to make it more specific.

Line 20: I do think the last paragraph of the Discussion provides some needed context that the 2021-2022 breeding season was wetter than average years and this could be noted in the Abstract. It felt like a big reveal at the very end of the paper that would have been nice to know up front.

R: Thanks for the suggestion. We agree with the reviewer and we have now introduced two important changes. In line 20 of the abstract we wrote: “This may be especially important in wet years, like the year when we conducted our surveys”. Additionally, under the Study Area description we added: “There are often inter-annual fluctuations in rainfall, alternating between wet and dry periods [18]. For example, mean annual precipitation between 2008 and 2019 was 177 mm (range: 65-315), and precipitation between July 2021 and June 2022 was 244 mm (38 % over the annual mean)” (Lines 131-133) 

Line 87: Perhaps ‘greater diversity of micro-habitats’ instead of ‘more’

R: Changed.

Figure 1: Maybe it’s just the resolution of the image that I am seeing in the document, but the different dot sizes are not very apparent and I’m wondering if size is worth including in the Figure without a legend indicating the magnitude of variation. In the caption, does the color refer to the latent spatial random effect (meaning more plovers in the red dots and less plovers in the blue dots)? I am not sure how to interpret “spatial random error”, but I would suspect you mean to say the value of the spatial effect at each site (positive or negative). It’s also interesting that there seems to be an east to west effect where there were more plovers in the eastern sites and less in the western sites. I’m wondering if the authors can hypothesize why there seems to be some clustering almost split in the middle of the surveyed sites.

R: We changed Figure 1 so that points do not include spatial random effects and improved our explanation of the legend: “Vilama wetlands. Each dot represents a surveyed peatland”. Because the interpretation of random spatial errors can be tricky for non-occupancy modelers we have decided to move figure 1 to the appendix and replace it with a figure where we only show the spatial location of surveyed peatlands.

We appreciate the reviewer’s inquiry about interpreting spatial random errors. These are not latent occupancy states and therefore they do not tell us anything in terms of the likelihood of a patch being occupied. Depending on the values of other covariates in the model, two response values (i.e., the latent occupancy z as well as occupancy probability psi) at sites with similar values of the random effect may or may not be similar. If the covariates have similar values, they will be very similar. If not, they could be different, which depends on the estimated magnitudes of the covariate effects relative to the spatial random effect values. 

We doubt there is a strong precipitation gradient in only 40 km and peatlands are fed by several mountains, therefore there is not an obvious explanation we feel comfortable speculating on. However, a plot of mean NDWI mean (our main predictor) against longitude in UTM coordinates does show some correlation between NDWI and space (with more peatlands showing higher NDWI values to the west). We believe these spatial pattens we observe (both for the random effects and partially for some of the covariates) support the incorporation of space to the models.

Fig 1. Mean NDWI plotted against UTM easting. In blue, peatlands where plovers were detected, in red peatlands where plovers were not detected. The plot shows some correlation between space and NDWI.

Line 126-130: Related to my previous comment, it would be beneficial to add how precipitation (and maybe temperature) during the study year compared to these general descriptions of climate conditions.

R: As described in our response to the reviewer's second comment, we added a sentence describing precipitation means and variation over the years before the study (Lines 131-133)

Line 228: Rather than ‘basically the same’, state that the covariates on occupancy were the same as the 12 models with constant detection probabilities.

R: Changed.

Line 230: What coordinates were used for the spatial random effect? Was it the centroid of each peatland, the starting point, or something else? The centroid would seem to be the least affected by the size of the peatland.

R: This is an important comment and we have now clarified this in lines 235-237. “To estimate random effects in the spatial models we used an exponential covariance function, where UTM coordinates were defined at the centroid of each peatland”

Line 234: Change ‘make’ to ‘made’

R: Changed.

Line 260: Easier to fit in what way?

R: We usually say a model is easy to fit if it does not require large computational effort and reliably converges to a solution. To make this clear now the sentence reads: “easier to converge.” (line 264)

Reviewer 4: This paper presents new data on factors that affect peatland use by a little-studied species, the Diademed Plover. The study is overall well-executed, and the statistical analyses are appropriate for the study system and research questions. I do not have any major concerns, and the authors have done a commendable job addressing previous reviewer questions. My only general question is to ask the authors how variable was grazing pressure among each peatland? Were there some peatlands that, for example, experienced zero grazing pressure, while others experienced heavy grazing pressure? And are counts of vicunas and llamas during each survey sufficient to characterize the grazing pressure experienced throughout the course of the season? I wonder if an effect could be detected if grazing pressure were measured using different methods, or if there were a wider range of grazing pressures present within the study system.

R: 32 out of 40 peatlands were characterized based on at least three visits and, as we wrote in lines 267-269 there was wide variation in grazing pressure (range: 0-9.79 sheep equivalents/ha, median: 1.78). Below we include an exploratory plot of grazing pressure that shows several peatlands have values near zero. In blue, peatlands where plovers where detected, in red peatlands were plovers where not detected.

Fig2. Grazing values (sheep equivalents/ha) for the 40 peatlands surveyed.

Minor editorial comments given below:

Abstract

Lines 7-10: These two sentences are somewhat repetitive and can be condensed.

R: We deleted the second sentence and the previous now reads: “We hypothesized that Diademed Plovers occupy preferably larger and more humid peatlands, and avoid peatlands used for grazing by llamas and vicuñas, which may trample vegetation and nests” (Lines 7-9).

Introduction

Lines 74-76: This sentence is somewhat awkward and should be restructured.

R: We re-wrote the sentence and it now reads: “For species like the Diademed Plover, using models based on occurrence (presence or presence-absence) can help identify factors affecting occupancy and also provide a cost-effective method to assess population status based on occupancy rather than abundance”. Lines 74-76.

Paragraph lines 74-82: This paragraph provides some general discussion of modeling occurrence/occupancy. It could use another sentence or two at the end to explain the relevance of this background information to the current study, perhaps by explaining the appropriateness of the study system for this type of occupancy model.

R: We added the following sentence before the end of the paragraph: “Site-occupancy models are the preferred option for habitat specialists, especially those not found in all patches and occuring at relatively low densities” (lines 80-81).

Line 95: coastal habitats?

R: Added. Thanks for the suggestion.

Lines 105-107: These two sentences are somewhat repetitive and can be condensed.

R: This was a single sentence in the original manuscript, but one of the reviewers asked to write separate sentences for hypotheses and predictions. Thus, we are leaving it as it now stands.

Methods

Line 137: What are these types?

 R: We added the features of these peatlands after the colon, but for further clarification and to more explicitly link this to the work of Izquierdo et al., we re-wrote the sentence as:

“Peatlands at Vilama can be classified as types 1 and 2 according to Izquierdo et al. [45]: they are generally dominated by cushion plants of the genus Distichia and Oxychloe and are located in relatively steep areas above 4000 m.a.s.l” (lines139-141).

Line 155: included

R: Corrected.

Line 172: encompassing

R: No changes made. The sentence is grammatically correct.

Results

Lines 276-278: The first half of this sentence could go to Methods section, paragraph lines 211-221 where this correlation is discussed. Also, I’m not sure why the plot of detections vs. NDWI min is included in Figure 3, given that this variable was discarded from the occupancy models due to correlation with NDWI mean.

R: We agree with the reviewer. We have now removed the correlation between the two indices from results and moved it to methods. The sentence in methods now reads: “NDWI min (the minimum NDWI min recorded over three years) and the range of NDWI (max NDWI − min NDWI) were both correlated with NDWI mean (r = 0.96 and r = -0.57 respectively, S1 appendix). We decided to use NDWI mean as a more reliable index of the long term moisture dynamics of peatlands” (lines 217-221).

For the reasons mentioned in the last part of the previous sentence we kept NDWI mean for our models but we consider relevant that the absolute driest peatlands (lowest NDWI values) are those more likely to be occupied.

Discussion

Line 315: Snowy Plovers

R: Corrected

Line 320: Oystercatcher

R: Corrected

Line 325: Plovers

Line 330: similarly-sized

R: We re-wrote the sentence and it now reads: “Other plovers of similar size” (line 333)

Line 340: Dunlins and Wilson’s Snipes

R: Corrected.

Line 341: “…shorebirds that show no defense against cattle like the Diademed Plovers” – wording is somewhat unclear. I suggest either omitting, or rewording as “shorebirds that, like the Diademed Plovers, show do defense…”

R: We followed the reviewer’s suggestion. The sentence now reads: “…shorebirds that, like Diademed Plovers, show no defense against cattle but at sites where livestock graze year-round.” (lines 344-345)

Lines 369-371: Which could lead to a positive association with NDWI in drier years – perhaps important to note.

R: We rewrote the sentence and it now reads: “In drier years, small peatlands may become unsuitable (no running water) and peatlands that were too humid for Diademed Plovers in 2021–2022 may become suitable, reversing the relation with NDWI” (lines 373-375)

---

## [Editor Report · Decision Letter 2]

31 May 2024

Habitat occupancy of the threatened Diademed Plover (Phegornis mitchellii) is not affected by llama grazing or peatland size, but declines with peatland humidity

PONE-D-24-00544R2

Dear Dr. Pietrek,

We’re pleased to inform you that your manuscript has been judged scientifically suitable for publication and will be formally accepted for publication once it meets all outstanding technical requirements.

Kind regards,

Vitor Hugo Rodrigues Paiva, Ph.D.

Academic Editor

PLOS ONE